

# Buried treasure in a public repository: Mining mitochondrial genes of 32 annelid species from sequence reads deposited in the Sequence Read Archive (SRA)

Genki Kobayashi

Research Center for Creative Partnerships, Ishinomaki Senshu University, Ishinomaki, Miyagi, Japan

Corresponding author
Genki Kobayashi,
genkikobayashi5884@gmail.com

## ABSTRACT

**Background**. The mitochondrial genomes (mitogenomes) of metazoans generally include the same set of protein-coding genes, which ensures the homology of mitochondrial genes between species. The mitochondrial genes are often used as reference data for species identification based on genetic data (DNA barcoding). The need for such reference data has been increasing due to the application of environmental DNA (eDNA) analysis for environmental assessments. Recently, the number of publicly available sequence reads obtained with next-generation sequencing (NGS) has been increasing in the public database (the NCBI Sequence Read Archive, SRA). Such freely available NGS reads would be promising sources for assembling mitochondrial protein-coding genes (mPCGs) of organisms whose mitochondrial genes are not available in GenBank. The present study aimed to assemble annelid mPCGs from raw data deposited in the SRA.

**Methods**. The recent progress in the classification of Annelida was briefly introduced. In the present study, the mPCGs of 32 annelid species of 19 families in clitellates and allies in Sedentaria (echiurans and polychaetes) were newly assembled from the reads deposited in the SRA. Assembly was performed with a recently published pipeline mitoRNA, which includes cycles of Bowtie2 mapping and Trinity assembly. Assembled mPCGs were deposited in GenBank as Third Party Data (TPA) data. A phylogenetic tree was reconstructed with maximum likelihood (ML) analysis, together with other mPCGs deposited in GenBank.

**Results and Discussion**. mPCG assembly was largely successful except for *Travisia forbesii*; only four genes were detected from the assembled contigs of the species probably due to the reads targeting its parasite. Most genes were largely successfully obtained, whereas atp8, nad2, and nad4l were only successful in 22–24 species. The high nucleotide substitution rates of these genes might be relevant to the failure in the assembly although nad6, which showed a similarly high substitution rate, was successfully assembled. Although the phylogenetic positions of several lineages were not resolved in the present study, the phylogenetic relationships of some polychaetes and leeches that were not inferred by transcriptomes were well resolved probably due to a more dense taxon sampling than previous phylogenetic analyses based on transcriptomes. Although NGS data are generally better sources for resolving phylogenetic relationships of both higher and lower classifications, there are ensuring

needs for specific loci of the mitochondrial genes for analyses that do not require high resolutions, such as DNA barcoding, eDNA, and phylogenetic analysis among lower taxa. Assembly from publicly available NGS reads would help design specific primers for the mitochondrial gene sequences of species, whose mitochondrial genes are hard to amplify by Sanger sequencing using universal primers.

## INTRODUCTION

The mitochondrial genomes (mitogenomes) of metazoans generally include the same set of 13 protein-coding genes (PCGs), namely, atp6, atp8, cox1–3, cytb, nad1–6, and nad4l. The mitochondrial PCGs (mPCGs) are homologous and thus the nucleotide sequences of mitochondrial genes flanked by conservative regions are therefore easily compared using sequences obtained with universal primers (*e.g.*, the cox1 gene; *Folmer et al., 1994*). In addition, such gene sequences are applied to various molecular techniques, such as DNA barcoding, eDNA, and phylogenetic analyses. However, the mitochondrial gene sequences of some lineages are hard or impossible to amplify using universal primers with standard protocols due to nucleotide substitutions in conservative sites (*Carr et al., 2011*; *Sun, Kupriyanova & Qiu, 2012*) and the insertion of group II introns between the annealing sites of universal primers (*Bernardino et al., 2017*; *Kobayashi, Itoh & Kojima, 2022c*). Such problems can be resolved by adjusting conditions for polymerase chain reaction (PCR) and/or designing more specific primers for a target group. Also, the mitochondrial gene sequences in sequence reads generated by next-generation sequencing (NGS), such as genome skimming and RNA sequencing (RNA-seq), can help in determining the nucleotide sequences of such lineages. Genome skimming is applicable to ethanol-fixed specimens that are not suitable for RNA-seq, which requires high-quality and specifically fixed samples.

The NGS data openly available in the National Center for Biotechnology Information Sequence Read Archive (NCBI SRA) can also be useful sources for obtaining the mitochondrial genes of species whose mitogenomes are not yet publicly available. RNA-seq and genome skimming are powerful tools for resolving phylogenetic relationships among higher taxa (*Kocot et al., 2011*; *Kocot et al., 2017*; *Weigert et al., 2014*; *Laumer et al., 2019*; *Ma et al., 2019*; *Martín-Durán et al., 2021*; *Tan et al., 2021*; *Taite et al., 2023*). Consequently, NGS data are accumulating for several phyla to elucidate inter-familial relationships that are hard to be resolved with limited genes. However, the annotations of each assembled gene used for the phylogenetic analyses are often not available in public databases (*e.g.*, GenBank) and are not readily used by other researchers. A recent study developed a pipeline "mitoRNA" for assembling mitochondrial genes from RNA-seq reads (*Forni et al., 2019*). This pipeline includes two steps for assembling mitochondrial genes from transcriptome data: (1) mapping all transcriptome reads on reference mitogenomes with Bowtie2 (*Langmead & Salzberg, 2012*); (2) assembling mapped reads with Trinity (*Grabherr et al., 2011*). The resultant contigs are then used as references for successive

iterations of these steps automatically. Newly assembled mitochondrial PCGs from public NGS data enable analyses with a wider taxon sampling than those solely based on published mitogenomes.

Mainly, two types of NGS data, transcriptomes and genomic data, would be a candidate for the assembly of mitochondrial genes. A lot of the transcriptome data of Annelida, one of the major components of marine benthos, are available in the SRA; searching with the keywords "Annelida" and "RNA-seq" yielded 2,162 records in 2022. Furthermore, the number of deposited transcriptomes has been increasing as the search with the same words returned 2,524 records in October 2023. In addition to the application of transcriptomes to evolutionary developmental biology, the transcriptomes of various annelid lineages are obtained to infer deep phylogenetic relationships (*Weigert et al., 2014*; *Andrade et al., 2015*; *Struck et al., 2015*; *Helm et al., 2018*; *Erséus et al., 2020*; *Martín-Durán et al., 2021*; *Tilic et al., 2022*) since the phylogenetic relationships of higher taxa in annelids were not sufficiently resolved in phylogenies based on several genes. Transcriptomes are also used for the phylogenetic analyses of interfamilial relationships of closely related families and intrafamilial relationships of annelids (*Lemer et al., 2015*; *Novo et al., 2016*; *Anderson et al., 2017*; *Stiller et al., 2020*; *Tilic et al., 2020b*; *Shekhovtsov et al., 2022*). Also, there has been an increase in the number of studies that employ genome skimming for phylogenetic analyses based on mitogenomes in recent years (*Richter et al., 2015*; *Bernardino et al., 2017*; *Zhang et al., 2018*; *Zhao et al., 2022*; *Struck et al., 2023*). Although the publicly available genomic data of annelids are still limited, those of annelids are rapidly accumulating led by large projects, *e.g.*, the Darwin Tree of Life Project (https://www.darwintreeoflife.org/), which aims to sequence genomes of all known species of eukaryotes in Britain and Ireland.

In the present study, the mPCGs of 32 annelid species of 19 families of clitellates and allies in Sedentaria (echiurans and some polychaetes) were newly assembled from the NGS reads, mainly obtained by transcriptome analysis (*Erséus et al., 2020*), deposited in the NCBI SRA database. Also, the phylogenetic analyses based on 13 mPCGs are conducted using clitellates and allies.

## MATERIALS & METHODS

Although the paraphyletic status of "Polychaeta" has been recognized for decades, the higher classification of Annelida is not yet stable (for recent reviews, see *Weigert & Bleidorn, 2016*; *Struck, 2019*). There has been notable progress in the classification of Annelida for the last decade and a half. Errantia and Sedentaria were resurrected (*Struck et al., 2011*) and a clade name Pleistoannelida was proposed for Sedentaria + Errantia (any of which were without a taxonomic rank) (*Struck, 2011*). *Andrade et al. (2015)* rejected Pleistoannelida mainly because they considered changes in the phylogenetic positions of several lineages shown by *Andrade et al. (2015)* made the original definition of Pleistoannelida inappropriate. Later, *Struck (2019)* adjusted the definition of Pleistoannelida to incorporate the entities of Errantia and Sedentaria shown by recent studies. *Rouse, Pleijel & Tilic (2022)* followed the view regarding Pleistoannelida in *Andrade et al. (2015)* although the changes in *Struck (2019)* were not cited in *Rouse, Pleijel & Tilic*

(2022). *Weigert & Bleidorn (2016)* proposed Palaeoannelida for the clade Magelonidae + Oweniidae. Recently, the considerably revised classification for Annelida was proposed in a book by researchers mainly working on polychaetes (*Rouse, Pleijel & Tilic, 2022*), including the redefinition of the class Polychaeta. Polychaeta sensu *Rouse, Pleijel & Tilic (2022)* comprises Errantia and Sedentaria and thus it includes leeches and oligochaetes but not early branching lineages that are the members of conventional polychaetes (*e.g.*, Magelonidae, Oweniidae, Chaetopteridae, and Amphinomidae). The common name ''polychaetes'' for Polychaeta sensu *Rouse, Pleijel & Tilic (2022)* therefore includes leeches and oligochaetes but not magelonids, oweniids, chaetopterids, or amphinomids. This situation is different from the conventional usage of polychaetes and can lead to confusion when using the word ''polychaetes'' (it is avoided when using ''pleistoannelids'' for Errantia + Sedentaria and polychaetes for conventional ''Polychaeta''). Furthermore, the taxonomic grouping of clitellates is also controversial: *Rouse, Pleijel & Tilic (2022)* proposed the order Oligochaeta, consisting of Aeolosomatidae, *Hrabeiella*, and the suborder Clitellata, within Polychaeta sensu *Rouse, Pleijel & Tilic (2022)*, whereas Clitellata was conventionally higher than Oligochaeta (see a recent proposed order-level classification of oligochaetes; *Schmelz et al., 2021*). For these reasons, further discussion on the classification of Annelida would be required to establish a stable and widely accepted classification of Annelida in the future. In the present study, common names (not as formal taxonomic groupings) polychaetes, oligochaetes, and clitellates are used to indicate the conventional grouping (*i.e.,* not polychaetes, *etc.* in *Rouse, Pleijel & Tilic, 2022*) to avoid confusion explained above.

The paired-end raw reads of 32 annelid species of Sedentaria were downloaded from the SRA with fasterq-dump v3.0.0 (Table 1). Adapter trimming and quality filtering ($Q > 30$) were performed using fastp v0.20.1 (*Chen et al., 2018*). The first five million reads were extracted from filtered reads with the ''head'' command of SeqKit v0.12.0 (*Shen et al., 2016*) to save computation time of assembly since the results of a preliminary assembly using all reads (32 million) of *Travisia forbesii* did not differ between assembly using five million reads. A pipeline mitoRNA (*Forni et al., 2019*), which uses Bowtie2 v2.4.5 (*Langmead & Salzberg, 2012*) for mapping reads on references and Trinity v2.14.0 (*Grabherr et al., 2011*) for assembling mapped reads, was used to assemble mPCGs from the reads with default settings. The mitogenomes of *Abarenicola claparedi oceanica*, *Notomastus* sp., *Urechis unicinctus*, *Nais communis*, *Olavius algarvensis*, *Tubifex tubifex*, *Acanthobdella peledina,* and *Ozobranchus jantseanus* were used as references for mitoRNA. The mPCGs were searched from assembled contigs with the ''nhmmer'' command (*Wheeler & Eddy, 2013*) (using the--max option) in HMMER v3.3.2 (http://hmmer.org/) using HMM files constructed with the ''hmmbuild'' command based on the alignment files of annelid mPCGs (same dataset as *Kobayashi, Itoh & Kojima, 2022c*). Then, searched sequences were used as references for GetOrganelle v1.7.5.1 (*Jin et al., 2020*) and/or NOVOPlasty v4.2.1 (*Dierckxsens, Mardulyn & Smits, 2017*), which were used to assemble the complete length of each gene from the contigs generated by mitoRNA. The average coverage of each gene was checked with the ''bbmap'' command implemented in BBtools (https://sourceforge.net/projects/bbmap/). The obtained genes were deposited in DDBJ/EMBL/GenBank through DNA Data Bank of Japan as TPA (Third Party Data) (Table S1 for accession numbers of each gene sequence),

**Table 1** **Accession numbers of sequence reads downloaded from NCBI SRA.** Newly assembled mitochondrial protein-coding gene sequences with TPA accession numbers are listed in Table S1.

| Family | Species | SRR |
|---|---|---|
| Opheliidae | *Ophelina acumulata* | SRR10997422 |
| | *Thoracophelia mucronata* | SRR2017631 |
| Thalassematidae | *Bonellia viridis* | SRR2017645 |
| Trichobranchidae | *Trichobranchus roseus* | SRR11434466 |
| Arenicolidae | *Arenicola marina* | SRR2005653 |
| | *Abarenicola pacifica* | SRR10997426 |
| Scalibregmatidae | *Scalibregma inflatum* | SRR8799334 |
| Travisiidae | *Travisia forbesii*[a] | SRR9888046 |
| Aeolosomatidae | *Aeolosoma* sp. | SRR11559519 |
| Hrabeiellidae | *Hrabeiella periglandulata* | SRR10997424 |
| Randiellidae | *Randiella* sp. | SRR10997431 |
| Parvidrilidae | *Parvidrilus meyssonnieri* | SRR8799336 |
| Capilloventridae | *Capilloventer australis* | SRR8799324 |
| Phreodrilidae | Phreodrilidae sp. A | SRR10997437 |
| Naididae | *Albanidrilus* sp | SRR10997452 |
| | *Chaetogaster diaphanus* | SRR10997419 |
| | *Bathydrilus rohdei* | SRR8799332 |
| | *Olavius* sp. | SRR8799329 |
| | *Potamothrix* nr *heuscheri* | SRR10997432 |
| | *Rhyacodrilus pigueti* | SRR8799325 |
| Propappidae | *Propappus volki* | SRR5353250 |
| Enchytraeidae | *Grania simonae* | SRR10997449 |
| | *Enchytraeus crypticus* | SRR10997417 |
| Lumbriculidae | *Lumbriculus variegatus* | SRR8842488 |
| | *Kincaidiana* sp. | SRR10997445 |
| Branchiobdellidae | *Cirrodrilus suzukii* | SRR8842480 |
| | *Branchiobdella kobayashii* | SRR8799326 |
| | *Holtodrilus truncatus* | SRR8842481 |
| | *Bdellodrilus illuminatus* | SRR8842477 |
| | *Triannulata magna* | SRR8842482 |
| Cylicobdellidae | Cylicobdellidae sp. | SRR8842484 |
| Haemopidae | *Haemopis sanguisuga* | SRR10997447 |

**Notes.**
[a]Gene sequences were not used for phylogenetic analysis.

PRJDB14830 (BioProject), and SAMD00561016–SAMD00561048 (BioSample). The species and SRA accession numbers used in this study are summarized in Table 1.

The assembled mPCGs were aligned with the corresponding genes of part of Sedentaria (clitellates, echiurans, and part of polychaetes in Sedentaria) whose mitogenomes are deposited in GenBank and were used for phylogenetic analyses (Table 2) (88 OTUs). Siboglinidae was used as an outgroup. This subset was selected since the mitogenomes of some families of Sedentaria, such as Serpulidae, Fabriciidae, and Spionidae, show an extremely high nucleotide substitution rate in their mitogenomes (*Seixas et al.,*

*2017*; *Tilic, Atkinson & Rouse, 2020a*; *Sun et al., 2021*; *Ye et al., 2021*). *Travisia forbesii* was excluded from subsequent analyses since only four genes were successfully assembled. The mPCGs of *Capitella teleta* were searched by nhmmer using genome data downloaded from EnsembleMetazoa (https://metazoa.ensembl.org/index.html) and included in the phylogenetic analysis. The amino acid sequences were translated using invertebrate mitochondrial code with the "translate" command with the "-M" option (translate initial codon at the beginning to M) in SeqKit v0.12.0 (*Shen et al., 2016*). The amino acid sequences were aligned with MAFFT v7 (*Katoh & Standley, 2013*) using the G-INS-I option. The aligned amino acid sequences were checked manually. The "tranalign" command implemented in EMBOSS v6.6.0.0. (*Rice, Longden & Bleasby, 2000*) was used to align nucleotide sequences based on the aligned amino acid sequences. Ambiguous positions were excluded using trimAl (*Capella-Gutiérrez, Silla-Martínez & Gabaldón, 2009*) with the "gappyout" option. The maximum likelihood (ML) analysis was conducted for amino acid sequences with IQ-TREE v1.6.12 (*Nguyen et al., 2014*) using 1,000 ultrafast bootstrap (UFBoot) replicates. The interpretations of UFBoot values (ufBS) are different from the normal bootstrap values and ufBS ≥ 95% would be reliable (*Minh et al., 2021*). The NEXUS partition files were prepared to input sequence data into IQ-TREE. The best-fit substitution model for each gene was selected with ModelFinder (*Kalyaanamoorthy et al., 2017*), implemented in IQ-TREE (Data S1). FigTree v1.4.3 (http://tree.bio.ed.ac.uk/software/figtree/) was used to draw phylogenetic trees.

The pairwise genetic distances (*p*-distances) of aligned nucleotide sequences of each gene were calculated with the "distmat" command of EMBOSS (*Rice, Longden & Bleasby, 2000*).

A degenerated primer, modified from HCO2198 (*Folmer et al., 1994*), was designed considering the alignment of the cox1 gene (HCO-clitealli: 5′-CTTCNGGRTGNCCRAARAAYCA-3′), which can be used as a pair of LCO-annelid (*Kobayashi, Itoh & Kojima, 2022c*).

## RESULTS

### Assembly of mitochondrial protein-coding genes

mPCG assembly was largely successful except for *Travisia forbesii*; only four genes (cox1, cox2, cytb, and nad5) of this species were detected from assembled contigs. For 31 other species, cox1 and cytb were successfully detected in all of them (Table S1; partly assembled genes are listed in Table S2). Also, cox2, cox3, nad1, nad3, nad4, nad5, and nad6 were obtained in ≥ 29 species. On the other hand, atp8, nad2, and nad4l were only successful in 22–24 species. The medians of pairwise differences of genes among species were higher than 50% in atp8 and nad6 (Fig. 1).

### Phylogenetic analysis based on amino acid sequences

In a polychaete lineage, Travisiidae + Scalibregmatidae was monophyletic (ufBS = 100%) and sister to a clade including Arenicolida (Arenicolidae and Maldanidae; ufBS = 100%) and Terebellida (Pectinariidae, Alvinellidae, Ampharetidae, Terebellidae, Melinnidae, and Trichobranchidae; ufBS = 100%) (ufBS = 99%) (Fig. 2). The support value between families in Terebellida was high for the position of Pectinariidae, which was sister to the

**Table 2  Mitogenome sequences obtained from GenBank and used for phylogenetic analysis, in addition to newly assembled gene sequences.**

| Family | Species | GenBank accession No. |
| --- | --- | --- |
| Moniligastridae | *Drawida gisti* | MN539609 |
| | *Drawida japonica* | KM199288 |
| Megascolecidae | *Amynthas aspergillus* | KJ830749 |
| | *Metaphire vulgaris* | KJ137279 |
| | *Perionyx excavatus* | EF494507 |
| | *Tonoscolex birmanicus* | KF425518 |
| Lumbricidae | *Aporrectodea rosea* | MK573632 |
| | *Eisenia balatonica* | MK642872 |
| | *Lumbricus terrestris* | U24570 |
| Rhinodrilidae | *Pontoscolex corethrurus* | KT988053 |
| Naididae | *Limnodrilus hoffmeisteri* | MW732144 |
| | *Nais communis* | MW770354 |
| | *Olavius algarvensis* | LR992058 |
| | *Tubifex tubifex* | MW690579 |
| Acanthobdellidae | *Acanthobdella peledina* | MZ562997 |
| Haemadipsidae | *Haemadipsa crenata* | MW711186 |
| Hirudinidae | *Hirudo medicinalis* | KU672396 |
| | "*Hirudo nipponia*" | KC667144 |
| | *Hirudo verbana* | KU672397 |
| | *Poecilobdella javanica* | MN542781 |
| | "*Poecilobdella manillensis*" | KC688268 |
| | *Whitmania acranulata* | KM655838 |
| | *Whitmania laevis* | KC688269 |
| | *Whitmania laevis* | KM655839 |
| Erpobdellidae | *Erpobdella japonica* | MF358688 |
| | *Erpobdella testacea* | MT584166 |
| | "*Erpobdella octoculata*" | KC688270 |
| Glossiphoniidae | *Glossiphonia complanata* | MT872697 |
| | *Glossiphonia concolor* | MT628565 |
| | *Haementeria officinalis* | LT159848 |
| | *Hemiclepsis yangtzenensis* | MN106285 |
| | *Placobdella lamothei* | LT159849 |
| | *Placobdella parasitica* | LT159850 |
| Piscicolidae | *Codonobdella* sp. | MZ202177 |
| | *Piscicola geometra* | BK059172 |
| | *Zeylanicobdella arugamensis* | KY474378 |
| Ozobranchidae | *Ozobranchus jantseanus* | KY861060 |
| Travisiidae | *Travisia sanrikuensis* | LC677172 |
| Ampharetidae | *Auchenoplax crinita*. | FJ976041 |

| Family | Species | GenBank accession No. |
|---|---|---|
| | *Decemunciger* sp. | KY742027 |
| | *Eclysippe vanelli* | EU239687 |
| Alvinellidae | *Paralvinella sulfincola* | FJ976042 |
| Melinnidae | *Melinna cristata* | MW542504 |
| Trichobranchidae | *Terebellides stroemii* | EU236701 |
| Terebellidae | *Neoamphitrite affinis* | MZ326700 |
| | *Pista cristata* | EU239688 |
| | *Thelepus plagiostoma* | MW557377 |
| Pectinariidae | *Pectinaria gouldii* | FJ976040 |
| Arenicolidae | *Abarenicola claparedi oceanica* | LC707921 |
| Maldanidae | *Clymenella torquata* | AY741661 |
| Cepitellidae | *Notomastus* sp. | LC661358 |
| Opheliidae | *Armandia* sp. | LC661359 |
| Thalassematidae | *Urechis caupo* | AY619711 |
| | *Urechis unicinctus* | EF656365 |
| Siboglinidae | *Lamellibrachia luymesi* | KJ789163 |
| | *Sclerolinum brattstromi* | KJ789167 |

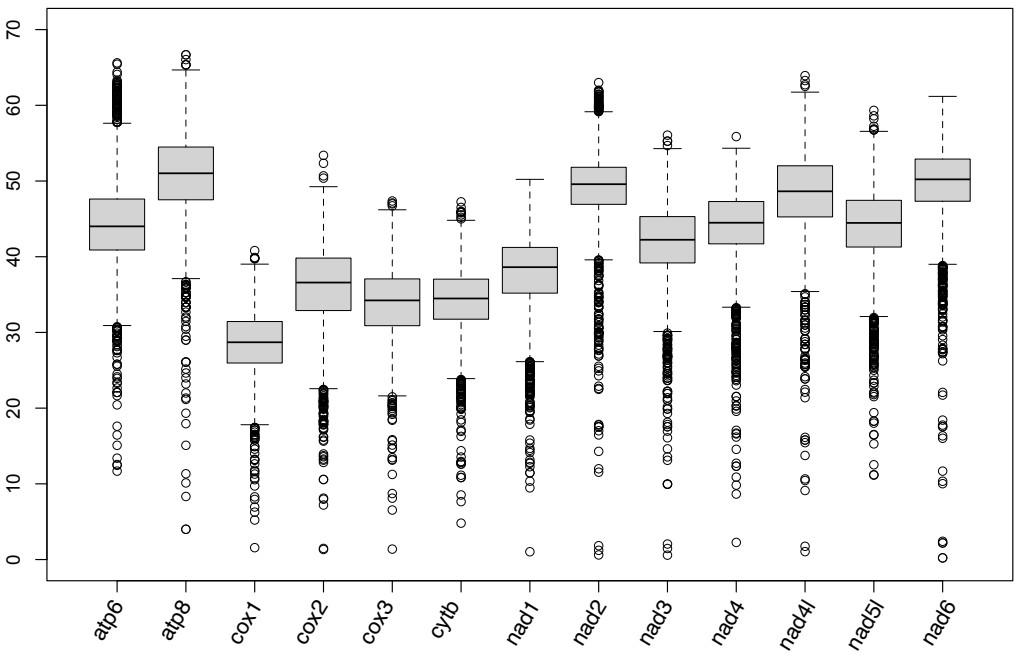

**Figure 1  Box plot showing the percentage of the pairwise genetic distances of protein-coding genes of mitogenomes.**

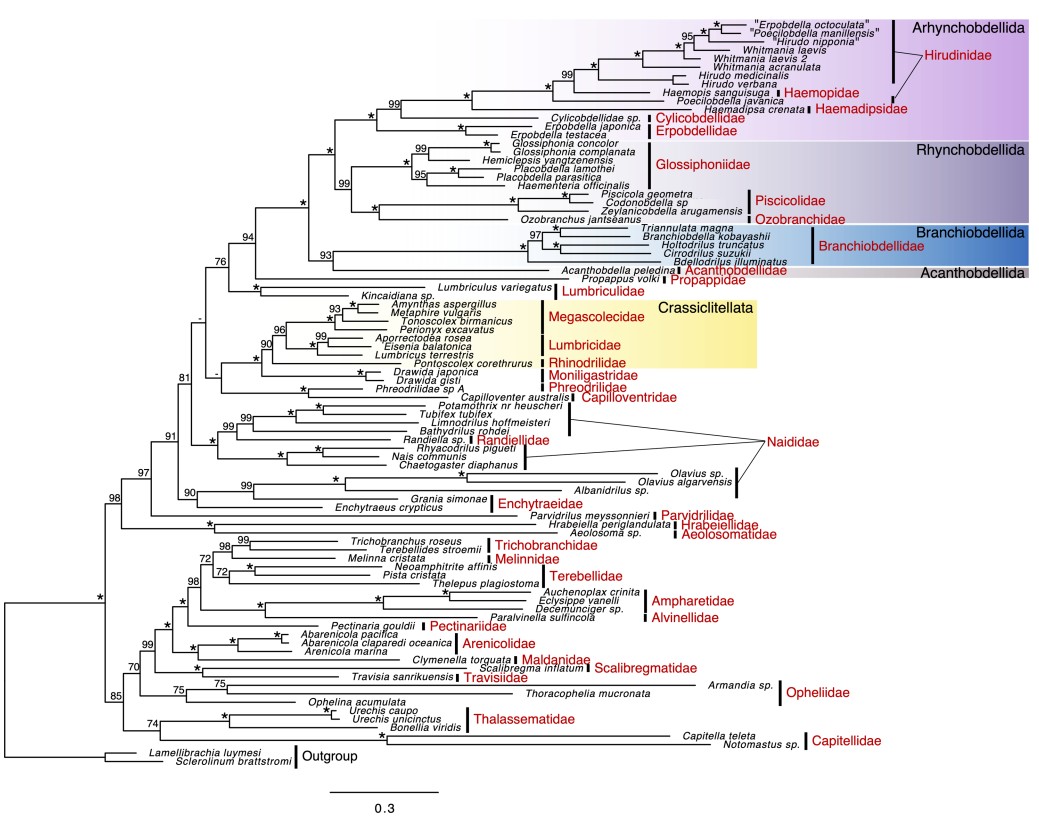

**Figure 2** Maximum likelihood phylogeny of a subset of Sedentaria based on the dataset, including amino acid sequences of 13 protein-coding genes (PCGs) after ambiguous positions were excluded (3,504 characters). Numbers above the branches represent the ultrafast bootstrap values (ufBS; ≥95% would be reliable; *Minh et al., 2021*). Asterisks indicate highly supported branches (ufBS = 100%).

remaining Terebellida (ufBS = 98%), Trichobranchidae + Mellinidae (ufBS = 98%), Ampharetidae + Alvinellidae (ufBS = 100%). Monophyletic Capitellidae (*Capitella teleta* and *Notomastus* sp.; ufBS = 100%) was sister to Thalassematidae (ufBS = 100%) although the support value was low (ufBS = 74%).

The sister relationship between monophyletic Aphanoneura (Aeolosomatidae and Hrabeiellidae; ufBS = 100%) and monophyletic clitellates (ufBS = 97%) was highly supported (ufBS = 98%). Parvidrilidae was sister to the moderately supported lineage of other clitellates (ufBS = 91%) although the support value was moderate (ufBS = 91%). The support values at deep nodes of oligochaetes and inside Naididae were largely low. A part of Naididae (*Olavius* spp. and *Albanidrilus*) was included in Enchytraeidae (*Enchytraeus crypticus* and *Grania simonae*) (ufBS = 90%). Randiellidae was nested within Naididae and the monophyly of this cluster was fully supported (ufBS = 100%). Monophyly of Phreodrilidae and Capilloventridae was fully supported, although the position of this clade was ambiguous. Propappidae was sister to the fully supported clade of leeches (ufBS = 94%), and Lumbriculidae was sister to this clade although the support value was low (ufBS = 76%).

UFboot support values in leeches were generally high. Branchiobdellida + Acanthobdellida (ufBS = 93%) was sister to Hirudinida (ufBS = 100%). Although Branchiobdellidae was monophyletic (ufBS = 100%), Branchiobdellinae (*Branchiobdella kobayashii* and *Cirrodrilus suzukii*) was not monophyletic: *B. kobayashii* was sister to Cambarincolinae (*Triannulata magna*) (ufBS = 100%) and *C. suzukii* was sister to *Holtodrilus truncatus* (ufBS = 100%). Each of Rhynchobdellida and Arhynchobdellida was monophyletic (ufBS = 99% and 100%, respectively). In Rhynchobdellida, Piscicolidae + Ozobranchidae clade (Oceanobdelliformes) (ufBS = 100%) was sister to Glossiphoniidae (ufBS = 99%). In Arhynchobdellida, Erpobdellidae (Erpobdelliformes) was sister to Hirudiniformes (Cylicobdellidae + Haemadipsidae + Haemopidae + Hirudinidae) (ufBS = 100%). Cylicobdellidae was sister to the other Hirudiniformes (ufBS = 99%). The sister relationship between Haemadipsidae and the Hirudinidae + Haemopidae clade (ufBS = 99%) was fully supported. *Poecilobdella javanica* (Hirudinidae) was sister to the highly supported lineage (ufBS = 99%) of Haemopidae + other hirudinids (ufBS = 100%), resulting in the paraphyletic nature of Hirudinidae.

## DISCUSSION

In the present study, the complete mPCGs of 11 families were registered in GenBank for the first time (Aeolosomatidae, Branchiobdellidae, Capilloventridae, Cylicobdellidae, Enchytraeidae, Hrabeiellidae, Randiellidae, Scalibregmatidae, Parvidrilidae, Phreodrilidae, and Propappidae). The mPCGs of the part of Sedentaria were well-assembled from the reads of RNA-seq or anchored hybrid enrichment (AHE) deposited in the SRA. Sequence reads generated by genome skimming were not used in the present study because of suitable datasets of the target group found in two previous studies using RNA-seq or AHE (*Phillips et al., 2019*; *Erséus et al., 2020*). Since genome skimming aims to skim the nucleotide sequences of the high copy fraction of the genome (*Straub et al., 2012*), prevalent organelle genome assemblers usually succeed in assembling the mitochondrial genome from the sequence reads generated by genome skimming. Assembly using the pipeline mitoRNA introduced in the present study may therefore be unnecessary for genomic data. The atp8, nad2, and nad4l genes showed high nucleotide substitution rates (Fig. 1). This may have resulted in the poor performances of nhmmer due to a divergence between query contigs and reference mitogenomes. These genes may be obtained when the mitogenomes of more closely related species or the species become available as the reference for assembly in the future. The low success rate, however, did not necessarily coincide with high nucleotide substitution rates (*e.g.*, the nad6 gene was assembled in 31 species although it shows a high substitution rate, which is similar to the nad2 gene). The atp8, nad2, and nad4l genes were also poorly assembled in Nereididae (Errantia) by a previous study on mitogenomic phylogeny of Nereididae (*Alves, Halanych & Santos, 2020*), which assembled mitochondrial genes from RNA-seq reads using Trinity. Unfortunately, the mPCGs of *Travisia forbesii* were poorly assembled probably because the RNA-seq targeted its apicomplexan parasite *Rhytidocystis* sp.(deposited as the transcriptome of *Rhytidocystis* sp. 1; Organism: *Rhytidocystis* sp. ex *Travisia forbesii*) and sequence reads derived from *T. forbesii* may be scarce.

In the present study, phylogenetic relationships in polychaetes and leeches that were not inferred by transcriptomes were well resolved probably due to a more dense taxon sampling than previous phylogenetic analyses based on transcriptomes or mitogenomes, enabled by using both mitogenome data and mitochondrial genes assembled from NGS data. Scalibregmatidae and Travisiidae were recovered as sister lineages, corroborating the previous studies based on phylogenetic analyses based on several genes (*Paul et al., 2010*; *Martínez, Di Domenico & Worsaae, 2013*; *Martínez, Di Domenico & Worsaae, 2014*; *Law, Dorgan & Rouse, 2014*). Although the family Haemopidae was recognized as a valid taxa in a recent study (*Kvist et al., 2023*), *Haemopis* was nested within Hirudinidae in the present study, providing further support for the polyphyly of Hirudinidae suggested by phylogenetic analyses based on several nuclear and mitochondrial genes (*Phillips & Siddall, 2009*; *Tessler et al., 2018*).

Aphanoneura (Aeolosomatidae + Hrabeiellidae) was recovered as the sister clade to the remaining clitellates in the present analysis, which is consistent with the results of phylogenetic analyses based on transcriptomes by *Erséus et al. (2020)*. *Erséus et al. (2020)* stated a more exhaustive taxon sampling among the "polychaetous" groups (especially *Parergodrilus* and *Stygocapitella*) will be needed to conclude the sister of clitellates, as suggested by *Weigert & Bleidorn (2016)*. *Parergodrilus* and *Stygocapitella*, *i.e.,* the members of Parergodrilidae, are now shown to be closely related to Orbiniidae, not to clitellates (*Struck et al., 2015*). Although the phylogenetic positions of some other annelid families were not yet revealed, the present study also supports the sister relationship between Aphanoneura and clitellates.

Unfortunately, the phylogenetic positions of several lineages, which were well resolved in the phylogenetic analyses based on transcriptomes, were not resolved in the present study. For example, the phylogenetic positions of Naididae (*Olavius* spp. and *Albanidrilus*), *Randiella* sp., Lumbriculidae, and the Phreodrilidae + Capilloventridae clade were ambiguous. Although the Acanthobdellida and Hirudinida clade was sister to Branchiobdellida in *Phillips et al. (2019)* with full support, the relationship among Acanthobdellida, Branchiobdellida, and Hirudinida was not resolved in the present study. Also, the phylogenetic positions of Capitellidae, Thalassematidae, and Opheliidae, which were contentious in the recent phylogenetic studies focusing on the mitogenomes of Sedentaria (*Kobayashi, Itoh & Nakajima, 2022b*; *Kobayashi, Itoh & Nakajima, 2022a*) despite well-resolved phylogenetic relationships based on transcriptomes, were not revolved in the present study. Mitogenomic phylogeny can be conducted with a more dense taxon sampling than phylogeny based on transcriptomes, it should be noted that support values in mitogenomic phylogeny of the higher classifications (*Sun et al., 2021*; Fig. 1 in *Kobayashi, Itoh & Nakajima, 2022a*) are likely to be lower than phylogeny based on transcriptomes with similar taxon sampling (*Struck et al., 2015*; *Martín-Durán et al., 2021*) (although it is apparent considering the difference in the total number of loci).

Although the phylogenetic analysis based on genome-wide data is a powerful method for resolving phylogenetic relationships of both higher and lower classifications, analyses based on mitogenomes or each widely sequenced mitochondrial gene (*e.g.*, cox1) are still enduring needs in taxonomy and environmental assessments that do not need high

resolution but require a wide range of taxon sampling, such as DNA barcoding, eDNA, and phylogenetic analyses between closely related species. In particular, the reference sequences for species identification are in great demand for meta-analyses since the barcode coverage for marine invertebrates is still quite low; only 22% of species in the European Register of Marine Species (ERMS) list have been sequenced (*Weigand et al., 2019*). The gene sequences assembled in the present study include species whose mitochondrial genes are registered for the first time and they would be usable as the reference sequences for future DNA metabarcoding studies. The mitochondrial genes of some species that gave nucleotide substitutions in annealing sites of universal primers are hard or impossible to obtain gene sequences by such primers, possibly being one of the reasons for the delay in accumulating reference sequences. Although designing specific primers for lower taxa can successfully obtain gene sequences that are hard or impossible to amplify by universal primers (*e.g.*, *Sun, Kupriyanova & Qiu, 2012*; *Williams et al., 2017*), universal primers with degenerated bases are still useful for some annelids with high nucleotide substitutions in mitochondrial genes and uncertain phylogenetic relationships among genera and/or species since there are no promising specific primers for such annelids yet. *e.g.*, some genera in Capitellidae (*Jeong et al., 2018*; *Jeong, Soh & Suh, 2019*) and Fauveliopsidae (G Kobayashi, 2023, unpublished data). Mitochondrial gene assembly from the NGS reads of the representative species of such taxa may be effective in designing specific primers and obtaining target gene sequences of them.

## CONCLUSIONS

Mitochondrial protein-coding genes were newly assembled from the reads of RNA-seq or AHE deposited in the NCBI SRA of 32 annelid species of 19 families, using a pipeline mitoRNA. The dataset for phylogenetic analyses includes mitochondrial protein-coding genes obtained from mitogenomes and transcriptomes in the database, resulting in a dense taxon sampling of polychaetes and leeches compared with previous studies using either mitogenome or transcriptome. Although the support value between families was largely low in oligochaetes, some phylogenetic relationships in polychaetes and leeches were well resolved. The assembled mitochondrial genes from publicly available NGS reads would be useful for analyses that do not require high resolutions, such as DNA barcoding, eDNA, and phylogenetic analysis among lower taxa.

## ACKNOWLEDGEMENTS

I am grateful to two anonymous reviewers for their invaluable comments on the earlier version of the manuscript.

### Funding
This work was supported by JSPS KAKENHI (grant number JP22K15174). The funders had no role in study design, data collection and analysis, decision to publish, or preparation of the manuscript.

### Grant Disclosures
The following grant information was disclosed by the author:
JSPS KAKENHI: JP22K15174.

### Competing Interests
The author declares that he has no competing interests.

### Author Contributions
- Genki Kobayashi conceived and designed the experiments, performed the experiments, analyzed the data, prepared figures and/or tables, authored or reviewed drafts of the article, and approved the final draft.

### Data Availability
The obtained genes are available in DDBJ/EMBL/GenBank through DNA Data Bank of Japan as TPA data:

BR001822–BR001891, YAAB01000001–YAAB01000013, YAAC01000001–YAAC01000012, YAAD01000001–YAAD01000013, YAAE01000001–YAAE01000013, YAAF01000001–YAAF01000012, YAAG01000001–YAAG01000013, YAAH01000001–YAAH01000013, YAAI01000001–YAAI01000013, YAAJ01000001–YAAJ01000013, YAAK01000001–YAAK01000012, YAAL01000001–YAAL01000013, YAAM01000001–YAAM01000011, YAAN01000001–YAAN01000013, YAAO01000001–YAAO01000012, YAAP01000001–YAAP01000013, YAAQ01000001–YAAQ01000013, YAAR01000001–YAAR01000011, YAAS01000001–YAAS01000013, YAAT01000001–YAAT01000013, YAAU01000001–YAAU01000011, YAAV01000001–YAAV01000013, YAAW01000001–YAAW01000013, YAAX01000001–YAAX01000013, YAAY01000001–YAAY01000009, YAAZ01000001–YAAZ01000004, YABA01000001–YABA01000012, PRJDB14830, SAMD00561016–SAMD00561033, SAMD00561035–SAMD00561048.

### Supplemental Information
Supplemental information for this article can be found online at http://dx.doi.org/10.7717/peerj.16446#supplemental-information.

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
