# Peer review of "Buried treasure in a public repository: Mining mitochondrial genes of 32 annelid species from sequence reads deposited in the Sequence Read Archive (SRA)"

_PeerJ, doi:10.7717/peerj.16446_

## Round 0.1 · original submission · Major Revisions

This study utilizes data mining to find mito genomes which is a great idea and is something that will be used more and more. However, there should be more discussion on other data types aside from transcriptomes - data mining can come from many sources. There also needs to be a more robust discussion surrounding the usability of this data from a biological point of view. The discussion is lacking on what we have learned about annelid species from this study.

Reviewer 1 ·

Basic reporting

Dear Editor,
The manuscript "Buried treasure in public repositories ..." is interesting since it reassembles data already available in public data bases and analyses the mitochondrial protein coding genes from different groups of Sedentaria (Annelida). A phylogenetic analysis using these mt protein coding genes is provided and results are discussed in comparison with previous results based on different data sets (e.g. transcriptomes). In my opinion, the discussion would improve considerably if the evolution of relevant morphological features would be traced and discussed. Therefore, I suggest the authors extending the discussion and making a synthesis of the different evolutionary hypotheses made on the studied group so far, analyzing the changes in the evolutionary history of the main morphological traits, mainly those that have been used as diagnostic characters of the different groups.
Some more comments in the attached revised manuscript.
Sincerely

Experimental design

Fine

Validity of the findings

Fine but see general comments

Annotated reviews are not available for download in order to protect the identity of reviewers who chose to remain anonymous.

Reviewer 2 ·

Basic reporting

The manuscript by Kobayashi shows the potential of mining publicly available data sources to resolve persisting phylogenetic questions. The study is well conducted, and the results are solid. The phylogenetic reconstruction is actually one of the most sensible one I have seen using mitochondrial genes in Annelida. All in all, the manuscript reads also very well and is well structured. However, I have a few major and several minor comments, which should be addressed first before I can suggest publication of the manuscript. All of these comments should be easy to be addressed as they only require adding or rewriting parts.

Major comments:
1) While the manuscript describes the potential to obtain mitochondrial gene information from transcriptomic data and shows it is usefulness, it lacks a comparison to the alternative solution of genome skimming approaches, which can obtain entire mitochondrial genomes. In case of hard to amplify barcodes, genome skimming approaches might be more useful as the generation of transcriptomic data requires especially preserved material and hence is usually much more challenging to conduct than amplification-guided sequencing. Genome skimming approaches can use the same material as amplification-guided approaches and usually will provide the researcher not only with the mitochondrial genome, but also with the nuclear rRNA cluster. Hence, this option needs to be discussed in the manuscript and the advantages and disadvantages of the different approaches be objectively discussed. At present, the manuscript is too biased towards just using transcriptomic data.
See:
- Richter, S., F. Schwarz, L. Hering, M. Böggemann and C. Bleidorn (2015). "The Utility of Genome Skimming for Phylogenomic Analyses as Demonstrated for Glycerid Relationships (Annelida, Glyceridae)." Genome Biology and Evolution 7(12): 3443-3462.
- Trevisan, B., D. M. C. Alcantara, D. J. Machado, F. P. L. Marques and D. J. G. Lahr (2019). "Genome skimming is a low-cost and robust strategy to assemble complete mitochondrial genomes from ethanol preserved specimens in biodiversity studies." PeerJ 7: e7543.
- Bohmann, K., S. Mirarab, V. Bafna and M. T. P. Gilbert (2020). "Beyond DNA barcoding: The unrealized potential of genome skim data in sample identification." Molecular Ecology 29(14): 2521-2534.
- Tan, M. H., H. M. Gan, H. Bracken-Grissom, T.-Y. Chan, F. Grandjean and C. M. Austin (2021). "More from less: Genome skimming for nuclear markers for animal phylogenomics, a case study using decapod crustaceans." Journal of Crustacean Biology 41(2).
- Struck, T. H., A. Golombek, C. Hoesel, D. Dimitrov and A. H. Elgetany (2023). “Mitochondrial Genome Evolution in Annelida—A Systematic Study on Conservative and Variable Gene Orders and the Factors Influencing its Evolution” Systamtic Biology, Advance Access Publication, https://doi.org/10.1093/sysbio/syad023
2) In addition to genome skimming, given the amplitude of mitochondrial genomes and genes available now it is more promising to design species-specific primers instead of universal primers. This will also be cheaper and easier than to generate transcriptomic data. The generation of transcriptomic data requires that the RNA is preserved, which is for example not the case in samples preserved in ethanol.
3) The description of the group investigated needs to be more specific as substantial parts of Sedentaria are not investigated. Essentially, the group analyzed is Clitellata plus the closed polychaete relatives. These are all on one branch in the sedentarian tree of life. This should be made explicit in the text and especially in the abstract and Introduction. As there is no name for this group, it could be called “Clitellata and polychaete allies”. Moreover a reason should be given, why did the author not use representative of the other branches of Sedentaria such as Spionidae, Serpulidae, Siboglinidae, Orbiniidae, and Flabelligeridae as seeds.
4) I suggest that the author provides a table listing for each species the gene that was recovered by indicating the completeness of it. For example, 100% when it was fully recovered from Start to Stop codon or, e.g., 90%, 80%, 70% and so forth, when larger parts at the beginning, in between or the end are lacking. This would not only provide the information that a gene was recovered, but also how much of it.

Minor comments:
1) Lines 61-62: It is not clear if the introns target the binding site of the universal primers or if they are in the region between the two primers that is amplified? This is a difference as the former makes it impossible for the primer to anneal, while the latter would still make it possible, but the product is just longer than expected. Accordingly, in the latter case, the settings could be adjusted to allow the amplifications of longer fragments.
2) Lines 100-101: Delete “e.g., the redefinition of 101 Polychaeta, which corresponds to Pleistoannelida in Struck (2011).” and additional citations to the Rouse et al. 2022 citation.
First, Struck did not redefine Polychaeta as Pleistoannelida. He suggested the name completely new. Second, so far, the only suggestion for a new definition of Polychaeta as well as some other names in Annelida has happened only in the book by Rouse et al. 2022. Different peer-reviewed recent reviews have used different names. Please include the reviews by Weigert & Bleidorn (Weigert, A. and C. Bleidorn (2016). "Current status of annelid phylogeny." Organisms Diversity & Evolution: 1-18.) and Struck (Struck, T. H. (2019). 7.2 Phylogeny. Annelida Basal Groups and Pleistoannelida, Sedentaria I. G. Purschke, M. Böggemann and W. Westheide. Berlin/Boston, De Gruyter: 37-68.). This will give a more balanced view of the debate and not a one-sided one as presented by Rouse et al. 2022.
Accordingly, throughout the text, please refer to polychaetes by this colloquial name as it makes conversation easier but do it as Polychaeta. Polychaeta is a praphyletic group and hence a proper name should not be used.
3) Line 107: Why did the author not use all reads and restrict it to 5 million ones? It could maybe help with challenging genes. In any case, a reason should be provided.
4) Line 138: It is not obvious here if the author aligned the entire mitochondrial genomes or the corresponding genes from these genomes? This needs to be more specific.
5) Lines 138-140: It should be added that Siboglinidae was used as an outgroup.
6) Line 163: As the group comprises only a subset of annelids, the primer should refer to that subset and not annelid as a whole as this would be misleading. As there is no formal name for this group, it could be, for example, clitealli for clitellates plus allies.
7) Line 170: Table 1 just shows a list of the used species for the study, but not the results of the genes found. Hence, referencing here to Table 1 is not appropriate.
8) Description of the phylogenetic results: When describing the support for sistergroup relationships the author should mention both the BS values for the monophyletic groups (if applicable), which are sister to each other (e.g., Clitellata and Aphanoneura), as well as the one grouping these two together (e.g., the clade of Clitellata and Aphanoneura). These values together make the strong support for a sistergroup relationship. For example, for the sistergroup relationship of Clitellata and Aphanoneura, the one mentioned now is in principal only a support value for the monophyly of Clitellata, but not for the sistergroup relationship.

Typographical suggestions:
Line 27: “and part of polychaetes” instead of “and the part of polychaetes”
Line 29: Abbreviations should always be explained, in this case TPA.
Line 33: This section is both Results and Discussion, and hence this section should be called “Results and Discussion”.
Line 61: “the insertion of introns in the target” instead of “the insertion of the intron in the target”
Line 137: The last sentence is incomplete.
Line 138: “of part of Sedentaria” instead of “of the part of Sedentaria”.

Experimental design

The experimental design is good and sufficient.

Validity of the findings

The validity of the results is given.

Additional comments

I indicated major revision as major parts will be added.

---

## Round 0.2 · Major Revisions

You have done a great job at adding discussion points, however there are a few large concerns that were not properly addressed. I agree with the reviewer that you need a more critical review of different methods and they need to be integrated here - there are a lot of different data types available which mean there are lots of opportunities for data skimming. If this paper is to be packaged as a novel approach to acquiring mitochondrial sequence data, then the approach needs to be thorough.

Reviewer 2 ·

Basic reporting

The author addressed two of my four major points of criticism only supervisially and with minimal effort. My minor concerns are properly addressed. The discussion on polychaetes and Polychaeta is an accurate reflection and very well written. The author put a lot of effort into this new discussion. I appreciate that, but the same effort should have been put into addressing my major points of concern. Hence, the reseverations I have in these points still remain and hence I still suggest major revision. If these are not addressed properly and with care and detail I will not be able to accept the paper.

1.) I asked for a better representation of other methods in the manuscript. This does still not happen, just a cursory mentioning of them in Introduction is not sufficient. It already starts with the abstract with provides the impression that one can get only transcriptomic data from SRA database. This is not true as there are also tons of genome skimming data available. The author argues that the main focus of the paper is on transcriptomic data and therefore he restricted his discussion to this. However, as the paper heralds this approach as a new method to increase the amount of mitondrial sequence data, the author needs to discuss this in the light of all possible approaches to get such data. Why for example did he not target genome sequence data in SRA? What are the challenges? No all genome skimming data resulted in published mitochondrial data.
Moreover methods of data mining are only sensible when it can be expected that sufficient amount of data are generated to fill in the gaps. In annelids, in the last decade this meant transcriptomic data, but the landscape is changing rapidly and the generation of genomic data also for annelids is gaining more and more momentum. For example see Darwin Tree of Life webpage (https://tolqc.cog.sanger.ac.uk/index.html) using the search term "Annelida". It will provide 120 annelid species for which there are genomes published or ongoing just for that project. Genomic data for different sequencing depth will outrun transcriptomic data very fast very soon. Any paper advocating the advantage of transcriptomic data has to discuss these in the light of these development and show the advanatges why it will still be worth using this approach in 2-5 years. There should be a reason provided why it is not outdated already in 2 years.
I am not against the method, but as any scientific method it needs to be addressed and discussed having all possible methods in mind highlighting all its advantages, but also its disadvantages. Only such can the reader make a well-informed decision if it is worth to use the methods the own questions or not. It would in this respect also help to point how the same methodological princple can be applied to other, but similar data. This is all lacking in the moment. To be honest, as it is now the paper is not appealing to a broader audience as it provides a very restricted scope and outlook and many will find it outdated already while reading the abstract.
2.) The same line of critism is the basis of my second point of major critism. Given the massiev amounts of data generated now and the procedure shown by the author it might be more effective to design order-, family-, or genus-specific now. This will overcome the problems associated with the universal primers. Again this is a different approach to get quickly a lot of mitochondrial data than mining transcriptomic data. Here again much more detail in the discussion should be provided. What are the advantages of the transcriptomic approach over using specifically designed primers? How could the approach get integrated in such an effort to design specific primers? When is such such a more tailored specific amplification approach using specific primer instead of universal primers more promising than doing transcriptomics (e.g., preservation of samples, challenges in collecting them, needing only a few genes for the questions at hand, more cost-efficient for hundreds of species)?

Minor comments:
Line 27: "allies in Sedentaria (echiurans and polychaetes)" instead of "allies in Sedentaria (echiurans, clitellates, and polychaetes)" as the allies refers only to the non-clitellate annelids.
Line 107: "allies (echiurans, and some polychaetes) in Sedentaria" instead of "allies in Sedentaria (clitellates, echiurans, and some polychaetes)"
Line 143: "(it is avoided when using "pleistoannelids")." instead of "(it is avoided when using "pleistannelids")."

Experimental design

Nothing to add.

Validity of the findings

See my comments in the basic reporting.

---

## Round 0.3 · accepted · Accept

The author has addressed all concerns and has made the changes required. I have reviewed the manuscript and this paper is ready for publication.